# Parsimonious Gap-Filling Models for Sub-Daily Actual Evapotranspiration Observations from Eddy-Covariance Systems

Danlu Guo *[ID], Arash Parehkar, Dongryeol Ryu [ID], Quan J. Wang [ID] and Andrew W. Western

Department of Infrastructure Engineering, The University of Melbourne, Parkville, VIC 3010, Australia; aparehkar@student.unimelb.edu.au (A.P.); dryu@unimelb.edu.au (D.R.); quan.wang@unimelb.edu.au (Q.J.W.); a.western@unimelb.edu.au (A.W.W.)
* Correspondence: danlu.guo@unimelb.edu.au

**Abstract:** Missing data and low data quality are common issues in field observations of actual evapotranspiration ($ET_a$) from eddy-covariance systems, which necessitates the need for gap-filling techniques to improve data quality and utility for further analyses. A number of models have been proposed to fill temporal gaps in $ET_a$ or latent heat flux observations. However, existing gap-filling approaches often use multi-variate models that rely on relationships between $ET_a$ and other meteorological and flux variables, highlighting a critical lack of parsimonious gap-filling models. This study aims to develop and evaluate parsimonious approaches to fill gaps in $ET_a$ observations. We adapted three gap-filling models previously used for other meteorological variables but never applied to infill sub-daily $ET_a$ or flux observations from eddy-covariance systems before. All three models are solely based on the observed diurnal patterns in the $ET_a$ data, which infill gaps in sub-daily data with sinusoidal functions (Sinusoidal), smoothing functions (Smoothing) and pattern matching (MaxCor) approaches, respectively. We presented a systematic approach for model evaluation, considering multiple patterns of data gaps during different times of the day. The three gap-filling models were evaluated together with another benchmarking gap-filling model, mean diurnal variation (MDV) that has been commonly used and has similar data requirement. We used a case study with field measurements from an EC system over summer 2020–2021, at a maize field in southeastern Australia. We identified the MaxCor model as the best gap-filling model, which informs the diurnal pattern of the day to infill by using another day with similar temporal patterns and complete data. Following the MaxCor model, the MDV and the Sinusoidal models show comparable performances. We further discussed the infilling models in terms of their dependence on data availability and their suitability for different practical situations. The MaxCor model relies on high data availability for both days with complete data and the available records within each day to infill. The Sinusoidal model does not rely on any day with complete data, which makes it the ideal choice in situations where days with complete records are limited.

**Keywords:** actual evapotranspiration; latent heat; infill; data-driven; missing data; flux tower; data quality; maize field

## 1. Introduction

Actual evapotranspiration ($ET_a$) is an important component of the global water balance, accounting for about 62% of global precipitation over land [1]. Understanding and measuring $ET_a$ can provide useful information for various water resources management applications, such as for catchment water yield, urban water supply and irrigation management [2]. In the context of irrigated fields, $ET_a$ consists of both the evapotranspiration from crop surface and soil evaporation, which can take a total of 50–95% of surface irrigation water [3].

The eddy-covariance (EC) technique is considered to be one of the best techniques to obtain continuous, high-frequency field measurements of $ET_a$ [4]. The technique measures

sensible heat and latent heat fluxes, H and $\lambda E$, where the latter corresponds to $ET_a$. However, missing and low-quality data are commonly seen in EC-based measurements due to instrument malfunctions, power failures and unfavorable weather conditions [4–7]. These data gaps limit the utility of these datasets when complete $ET_a$ records are necessary, such as for water balance calculations. Therefore, effective gap-filling techniques are important to improve the completeness of $ET_a$ data obtained from EC systems.

Several approaches have been developed to infill gaps in the $\lambda E$ flux observations or the $ET_a$ observations directly, which have also been tested over a large range of field conditions including various climates zones and land cover types. Table S1.1 in the Supplementary provides a detailed summary of existing literature in these infilling approaches. These approaches largely rely on meteorological and/or other flux variables measured at the same location and over the same period as the variable to infill [7]. These gap-filling methods can be categorized into: (1) infilling $\lambda E$ (or $ET_a$) gaps with available records from neighboring time steps or with similar meteorological conditions [4,8–10], (2) building regression models between $\lambda E$ (or $ET_a$) and meteorological data, which sometimes also require further monitoring data for soil moisture and vegetation conditions such as leaf area index (LAI) [4,11–14]; and (3) predicting $\lambda E$ (or $ET_a$) with complex statistical models such as Kalman filter, multiple imputation or machine-learning algorithms developed based on meteorological conditions [11,12,14,15].

Although numerous gap-filling models were developed for $\lambda E$ flux and $ET_a$, they share a common limitation in the high model complexity and data requirement, highlighting a critical lack of parsimonious gap-filling models that operate only with the variable to infill itself. With the exception of two models, all other existing gap-filling approaches are dependent on relationships between $\lambda E$ flux or $ET_a$ and other driving variables, such as additional flux and meteorological variables. The two exceptions are the mean diurnal variation (MDV) [10] and the analogue period (AP) [16] methods in which gap-filling relies solely on the variable to infill itself. Thus, the applicability and performances of most of these gap-filling methods are highly dependent on the quality and availability of those additional variables. For example, the feasibility of such methods is limited when the required meteorological/flux data are also missing [7,16]. Further, the infilling may be affected by spurious relationships produced from changing meteorological conditions and/or outliers and low-quality records within the meteorological observations [4].

This study therefore aims to develop and evaluate parsimonious models to infill gaps in $ET_a$ observations derived from EC systems; we focus on parsimonious models that require only $ET_a$ itself, and thus having no reliance on data for other variables. Compared to the wide range of existing methods to fill data gaps in $ET_a$ and $\lambda E$ flux, the models presented here are easier to implement and are more applicable in data-limited situations. The simpler model structures also remove the dependence on the quality of measurements other than $ET_a$ (e.g., flux variables and meteorological data), which are often used as model predictors in existing infilling models. We adapt three parsimonious models that have never been used to infill sub-daily $ET_a$ data before. Two models are fitted to the observed diurnal patterns of $ET_a$ data in days with gaps, and one model utilizes days with complete $ET_a$ records to identify a matching temporal pattern to infill each day with gaps. These new models were compared with the mean diurnal variation (MDV) model, which is an existing parsimonious model that fills gaps in sub-daily data by averaging values recorded at the same time step within a short time window around the gap [10]. The MDV model has been widely used to infill gaps in flux variables [4,8,14,17].

We present a systematic approach to evaluate different infilling models considering multiple patterns of data gaps during different times of the day. We used a case study using field measurements from an EC system over summer 2020–2021 at a maize field in southeastern Australia. We discuss the relative performances of the four models along with their dependence on data availability, from which recommendations are made for different practical conditions. These gap-filling models and model recommendations presented will be highly valuable for improving the completeness and utility of $ET_a$ measurements from

EC systems in future studies. We made the R codes of all models evaluated in this study publicly available on GitHub https://github.com/DanluGuo/ETinfilling/blob/main/4_ETinfilling_Models_V2.R (accessed on 1 February 2022) along with example data.

## 2. Materials and Methods

The three new parsimonious gap-filling models, along with the existing model, MDV, were evaluated with field monitoring data from an EC system. The monitoring site and instrumentation is introduced in Section 2.1. The gap-filling models are introduced in Section 2.2. Section 2.3 describes the evaluation process, including (1) data resampling to represent typical missing/erroneous data at different times in a day; and (2) performance assessment and comparison for the four infilling models.

### 2.1. Monitoring Site for the Eddy-Covariance System and Data

To evaluate different $ET_a$ gap-filling models, we used monitoring data from an eddy-covariance system installed at a maize field over the 2020–2021 summer cropping season. The study field was within the Goulburn-Murray Irrigation District in southeast Australia (field centered at $-36.18S$, $145.04E$). The typical cropping season for summer maize in this region spans from November to May, while the study field was sown on 11 December 2020 and harvested on 23 April 2021. The field is located between temperate and arid steppe climate regions [18], with an annual mean rainfall of 447 mm, based on records at the closest public weather station (Kyabram, Australian Bureau of Meteorology #80091, 19 km away), from 1964 to 2021.

We continuously monitor the in-field weather condition alongside $CO_2$, $H_2O$ and sensible heat fluxes using an eddy-covariance system, between 19 December 2020 and 12 April 2021. We monitored the air temperature, solar radiation, relative humidity and wind speed at 2 meters' height from a standing weather station. The $CO_2$ and $H_2O$ fluxes were measured using an open path infrared gas analyser (LI-7500, Lincoln, NE. LI-COR, Inc.) and a three-dimensional (3D) sonic anemometer (CSATS3, Campbell Scientific Australia) as the core of the eddy-covariance system. All flux variables were monitored and recorded at 20 Hz frequency and subsequently processed and aggregated to 30-min interval with EddyPro® Software (Version 7.0) [19].

Figure 1 shows a photograph of the set of full monitoring equipment in field. The EddyPro® Software processes raw eddy covariance data to compute biospheric/atmosphereic fluxes of $CO_2$, $H_2O$ and sensible heat including applying raw data filtering, calibration, and other algorithms for calculating and correcting fluxes [20]. The remaining energy balance components were monitored with Kipp and Zonen CNR-1 radiometer, HFT3-L REBS soil heat flux plates and TCAV soil temperature thermocouples. The solar radiation was monitored at a 30-min interval, while air temperature, wind speed and relative humidity were monitored every 5-min; therefore, the $ET_a$ observations processed by EddyPro® was in a 30-min time step. The maize field is rectangular with cropping rows along the east–west gradient. Within the field, the weather station and the flux tower were placed next to each other at 10 m away from the northern field boundary and over 100 m away from the other three boundaries. Therefore, the vast majority of the target footprint was located at the southern side of the monitoring stations.

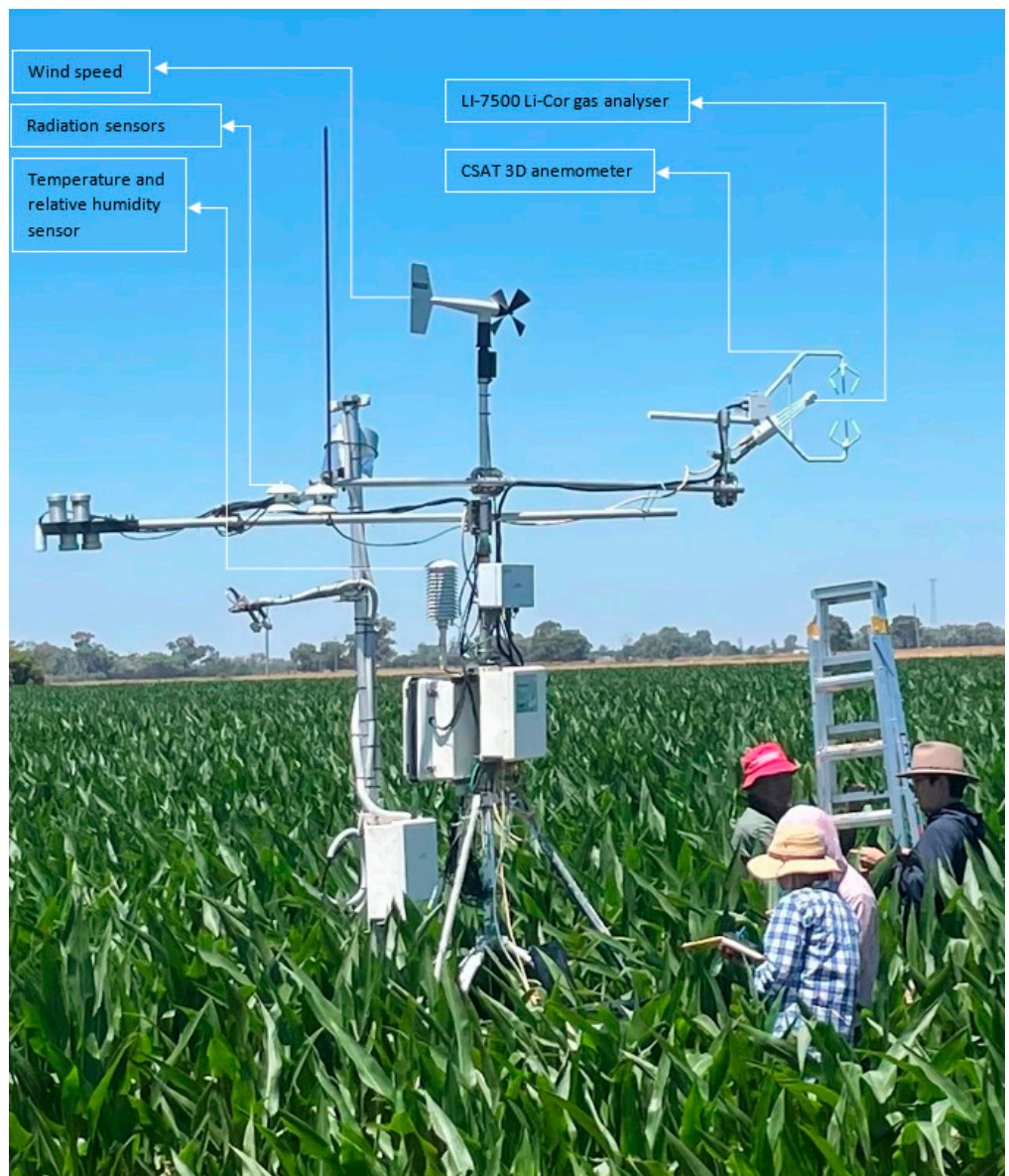

**Figure 1.** The eddy-covariance system and the weather station for monitoring fluxes and weather conditions at the study field. The purposes for different parts of the monitoring stations are labelled. Note that the CSAT 3D anemometer was measuring the 3D wind speed and direction, while the wind speed sensor on the top provided a second set of wind speed measures and the dominant direction within the horizontal plane for validation. The soil moisture down to 90 cm and the canopy temperature and NDVI were also monitored but observations were not used in this study.

Thirty-minute $ET_a$ data were estimated with EddyPro using the observations from the EC systems. The reference evapotranspiration ($ET_0$) data at corresponding time steps was estimated using the weather observations and the FAO-56 Penman–Monteith model [21]. The key quality issue for the $ET_a$ data occurred during periods when the wind was blowing from the north due to the limited fetch across the crop and a flux source footprint unrepresentative of the crop; therefore, the corresponding flux measurements may not be representative of the field, leading to inaccurate $ET_a$ estimates and poor energy balance closures. When wind was from southerly directions between 112.5 and 247.5 degrees (i.e., ESE to WSW), the sum of sensible and latent heat fluxes accounted for a median of 90% of available energy, suggesting a good energy closure (Figure 2). The gaps and potential errors in the $ET_a$ measurements prompted the need to develop effective gap-filling approaches.

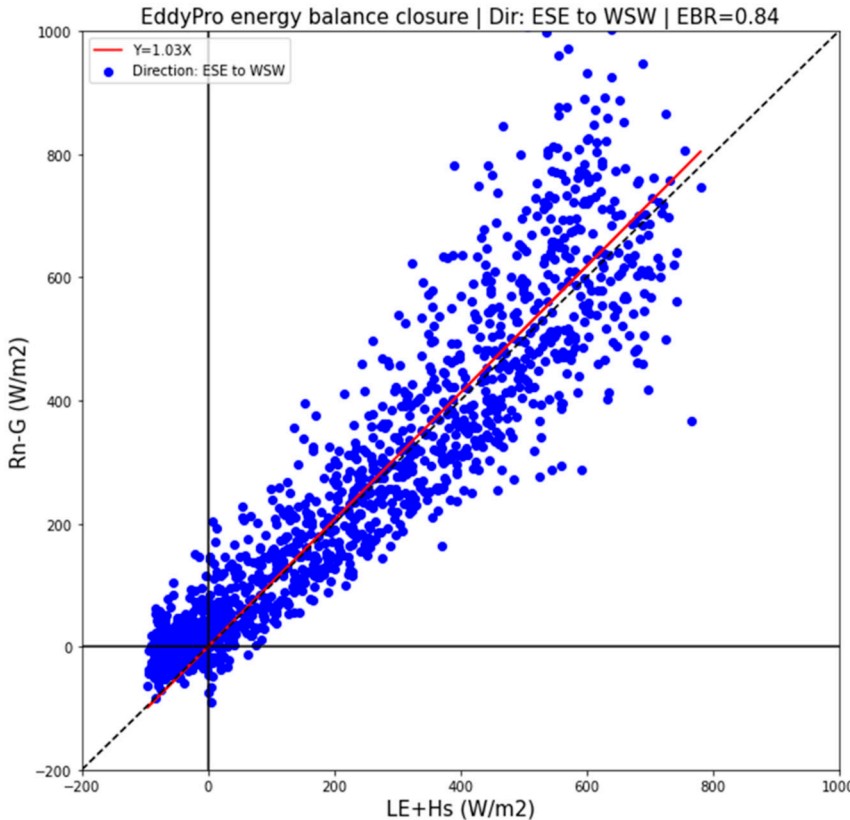

**Figure 2.** When wind directions are between 112.5 and 247.5 degrees (i.e., ESE to WSW), the heat fluxes (the sum of sensible and latent heat fluxes) accounted for a median of 90% of available energy, suggesting a good energy closure.

### 2.2. Three Gap-Filling Models for Sub-Daily $ET_a$

We adapted three models to infill gaps in sub-daily $ET_a$ and evaluated their performance along with a benchmarking model using our field observations. Suppose that each day within the observation period can be categorized by data availability as either a full day (FUL), a partial day (PAR), or a sparse day (SPA) as follows:

- Full day (FUL)—where data within the day is complete or mostly ($\geq$80%) complete;
- Partial day (PAR)—where part of the data within the day is missing but a substantial portion (30–80%) is still available;
- Sparse day (SPA)—where data within the day is mostly (>70%) missing/erroneous.

None of the three gap-filling models that we adapted and evaluated require data other than $ET_a$ records themselves. The models use the available records differently to fill gaps in the daytime 30-min $ET_a$ records within each PAR day (i.e., day to infill). The Sinusoidal model (*Daily sinusoidal functions of $ET_a$*) describes the diurnal patterns of $ET_a$ with a sinusoidal function of the time in a day, which have been widely applied to fill gaps in time-series of meteorological data, soil heat flux data, and even gaps in daily evapotranspiration [22–24]. The Smoothing model (*Daily smoothing functions of $ET_a$*) describes the diurnal patterns of $ET_a$ with polynomial functions of the time in a day, which is adapted from a common gap-filling approach for meteorological time-series [25]. The MaxCor model (*Daily temporal pattern matching for $ET_a$*) fills gaps in a day based on another day with complete records that is selected as having the most similar diurnal pattern with the day to infill. This model is conceptually similar to the analogue period (AP) model which fills a gap by searching the full dataset for an 'analogue period' that that has similar temporal patterns with the data surrounding the gap [16]. However, the implementation of MaxCor is much simplified as the search is based on a daily time-step, rather than the more flexible, user-defined time step as implemented in AP; specifically, MaxCor searches

for an 'analogue day', while AP allows any length of analogue period and recommends case-specific investigation to determine the optimal length. None of the three models have been used to infill sub-daily $ET_a$ data to our knowledge.

As a benchmark to the three abovementioned new models, we also included the mean diurnal variation (MDV) model in our evaluation, as this has been a widely used parsimonious gap-filling model for λE and carbon fluxes [4,8,14,17]. Similar to the three models introduced in this study, MDV requires the variable to infill as the only input. The MDV model was originally developed to fill gaps in laten heat flux observations [10]. The model fills any gap in sub-daily records by averaging values measured at the same time step on days adjacent to (both before and after) the gap, within a time window usually between 4 and 15 days. A shorter averaging period is considered insufficient to determine a reasonable mean value, while a longer average period might introduce errors due to potential non-linear impacts from other environmental variables. More details on the MDV model are included in its original paper [10]. For the model evaluation in this study, we implement MDV to infill any missing 30-min $ET_a$ records by the average of all values recorded at the identical 30-min time slot, within the adjacent 14 days (i.e., 7 days before and 7 days after the day where gap presents).

Common to all infilling models, a specific daytime period is defined for each day to infill, based on solar time angles estimated with the latitude and longitude of the study site and the ordinal dates within the record period, following Chapter 3 of the FAO-56 guidelines [21]. Across the season, the ranges of sunrise and sunset times are between 5:30 a.m. and 6:30 a.m. and 5:30 p.m. and 7 p.m., respectively; solar noon is between 12 p.m. and 12:15 p.m. Any $ET_a$ for times outside of daytime is treated as night-time $ET_a$ and assumed negligible. Any day that belongs to the SPA set is not filled because the available data is considered insufficient to be filled reliably. The threshold chosen to define the SPA set (having >70% of the 30-min $ET_a$ data missing) implies that a day can be infilled with a minimum of 15 out of 48 records available. This is a relatively low data requirement which could lead to unreliable gap filling. However, since our primary aim is to present and evaluate gap-filling models, a more important consideration in choosing the threshold was to enable a reasonable number of days remaining to be used for model evaluation (see Figure S1.1 in the Supplementary Materials for an assessment of data availability across the season). Details on how the FUL and PAR datasets were used for model evaluation is included in Section 2.3.

The three infilling models are detailed subsequently:

- *Sinusoidal—Daily sinusoidal functions of $ET_a$*: This model uses all available daytime 30-min $ET_a$ records on the day to be infilled (each day in the PAR set) to fit a sinusoidal function between $ET_a$ and time of the day, which has a period specific to that day. The fitted sinusoidal curve is then used to estimate all 30-min daytime $ET_a$ while infilling the missing time steps. We chose the sinusoidal function because of its simplicity and ability to represent the overall diurnal patterns of $ET_a$, which we concluded from a visual assessment of $ET_a$ for the FUL days within our records (days with >80% complete data, see details in Section 2.3.1). The sinusoidal function used takes the form of:

$$ET_{a,H} = Amp \times \sin\left(\frac{2\pi H}{P}\right) \tag{1}$$

Equation (1) describes the diurnal pattern of daytime 30-min $ET_a$ with the positive half of a sine curve. $H$ is the time since sunrise in decimal hours at the start of each 30-min slot (e.g., 1, 1.5, 2 h, and $H = 0$ at sunrise). Ideally, the full period of the sinusoidal curve, $P$, should be equal to twice of the daytime length (time between sunrise and sunset) of each day, which enables the representation of all daytime 30-min $ET_a$ with half of the sine curve where the daily peak occurs halfway between the sunrise and sunset. However, we found via a preliminary analysis on all FUL days that the daytime $ET_a$ peaks around 1:30 p.m., and the $ET_a$ diurnal patterns seem to follow only part of the half-sine curve (Figure 3). To represent these diurnal patterns in daytime $ET_a$ more accurately, the period $P$ in Equation

(1) is defined for each day as four times the hour difference between the sunrise and the hour of peak daytime $ET_a$ (1:30 p.m.), and the modelled $ET_a$ from Equation (1) after sunset of each day is zeroed; this adjustment of the sinusoidal function ensures that the model best resembles the observed asymmetrical diurnal patterns in $ET_a$. *Amp* is the only model parameter to be calibrated, which represents the amplitude of the sine curve; it is fitted by minimizing the sum of squared residuals from the available data on the day to be infilled.

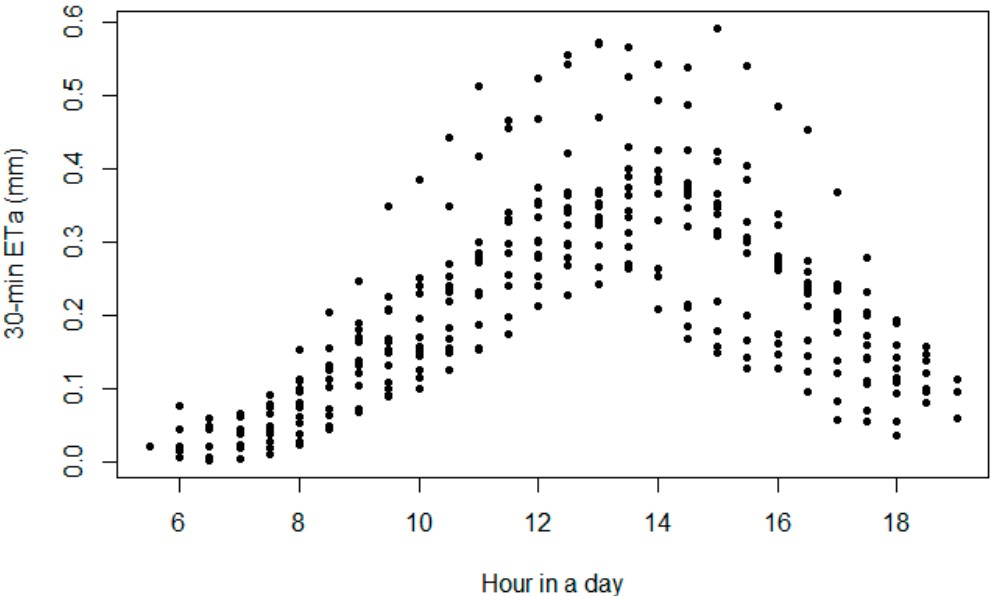

**Figure 3.** The diurnal patterns of 30-min daytime $ET_a$ across all days in the season with complete data (FUL days). The values of 30-min $ET_a$ generally peak around 1:30 p.m. and follow the shape of an incomplete half-sine curve.

- *Smoothing—Daily smoothing functions of $ET_a$*: This model uses all available daytime $ET_a$ data on the day to be infilled to fit a second-order polynomial smoothing function between $ET_a$ and time of the day. The fitted smoothing function is then used to infill $ET_a$ for the missing time steps. The second-order polynomial smoothing function takes the form of:

$$ET_{a,H} = AH^2 + BH + C \qquad (2)$$

In Equation (2), $H$ is the time since sunrise in decimal hours at the start of each 30-min slot; *A, B* and *C* are the model parameters to be calibrated.

- *MaxCor—Daily temporal pattern matching for $ET_a$*: For each day in the PAR set, this model first calculates the linear correlation (i.e., Pearson correlation coefficient) between the daytime $ET_a$ records in the current day and each day in the FUL set. The correlation calculation only considers the common timeslots where data are available in both the current day and each FUL day. Based on these correlations, the FUL day that has the maximum correlation with the day to infill is selected. Within this 'matching FUL day', all individual 30-min daytime $ET_a$ values ($ET_a\_FUL$) are divided by their sum ($ET_a\_tot\_FUL$) to calculate the proportions of 30-min $ET_a$ to the daily total, $ET_{a\_prop}$. This is described in Equation (3), where $H = 0, 0.5, 1, \dots, 24$, denoting the time since sunrise in decimal hours:

$$ET_{a-prop,H} = \frac{ET_a\_FUL_H}{ET_a\_tot\_FUL} = \frac{ET_a\_FUL_H}{\sum_{H=0}^{24} ET_a\_FUL_H} \qquad (3)$$

To infill the data gaps in the PAR day, we first estimate the daily total $ET_a$ of this day ($ET_a\_tot\_complete\_PAR$) by dividing the sum of all available $ET_a$ records ($ETa\_tot\_avail\_PAR$) by the sum of $ET_{a\_prop}$ values corresponding to these timeslots with available data. This

is described in Equation (4), where *Hpar* are the time since sunrise (decimal hours) for all timeslots with available records in the PAR day:

$$ET_a\_tot\_complete\_PAR = \frac{ET_a\_tot\_avail\_PAR}{\sum_{i \in Hpar} ET_{a-prop,i}} = \frac{\sum_{i \in Hpar} ET_a\_PAR_i}{\sum_{i \in Hpar} ET_{a-prop,i}} \tag{4}$$

We can then estimate each 30-min $ET_a$ value for the PAR day ($ET_a\_PAR$) with the estimated total $ET_a$ for the day ($ET_a\_tot\_complete\_PAR$) and all proportions of 30-min ETa, $ET_{a\_prop}$, which enables us to fill the $ET_a$ gaps (Equation (5)).

$$ET_a\_PAR_H = ET_a\_tot\_complete\_PAR \times ET_{a-prop,H} \tag{5}$$

We made the R codes which implement all four abovementioned models available on GitHub https://github.com/DanluGuo/ETinfilling/blob/main/4_ETinfilling_Models_ V2.R (accessed on 1 February 2022) along with the data used in this study.

### 2.3. Model Evaluation Process

Although the ultimate goal of the above-mentioned gap-filling models is to infill sub-daily data for days with partially missing data (i.e., the PAR set, as detailed in Section 2.2), our model evaluation was based on days within the FUL set only to understand the performance of individual infilling models. Specifically, we divided days within the FUL set into training and evaluation sets. We added artificial gaps to data (i.e., assign an NA value to some of the data) in the evaluation set to represent typical types of missing data from the field observations.

### 2.3.1. Classifying Daily Data Completeness

We first classified all days in our monitoring period into the FUL, PAR, or SPA sets by the completeness of data in each day, as highlighted by different colors in Figure 4.

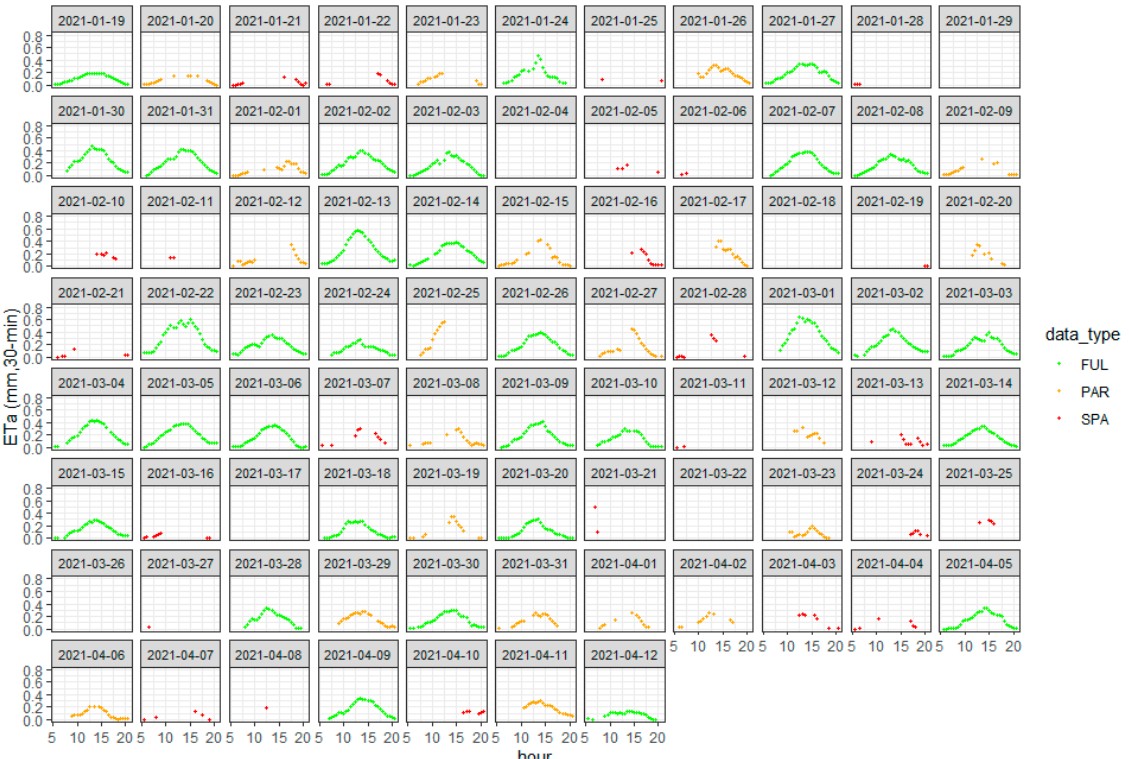

**Figure 4.** Classification of the completeness of 30-min $ET_a$ data for each day within the observation period. The colors differentiate days within the FUL, PAR and SPA sets. See the below text for the explanation of individual categories and their utility.

The data were then used as follows.

1.  The FUL set (green in Figure 4) contains days with complete/near complete (≥80%) records. These data will be further divided for training and evaluation of the four infilling models (Section 2.3.2).
2.  The PAR set (orange in Figure 4) contains days with partially complete (30–80%) records. These data were then used to summarize the typical patterns of missing data. We identified three typical patterns of missing data as:

    -   A: with most missing data in the morning (sunrise to 10 a.m.);
    -   B: with most missing during mid-day (10 a.m. to 3 p.m.);
    -   C: with most missing during afternoon (3 p.m. to sunset).

The days highlighted in red in Figure 4 are classified into the SPA set, where data for most (>70%) of the day were missing. As discussed in Section 2.2, these days are not recommended to be infilled because of significant lack of 'ground truth'.

### 2.3.2. Building the Training and Evaluation Datasets

Figure 4 identified 32 days within the FUL set, which were then randomly divided into:

1.  A training set (60%, 19 days); and
2.  An evaluation set (40%, 13 days).

The training dataset was used to represent days with complete data (the FUL set). Within the evaluation set, we added artificial gaps to each day to mimic each of the three typical patterns of missing data (A, B and C, Section 2.3.1). This led to three separate evaluation sets representing each type of missing data to be infilled with the four models. For each day in each evaluation set, data points corresponding to the gaps were held off (assigned as NA) and used for evaluating the performance of infilling models. For example, Figure 5 shows the training and evaluation sets that represent missing data Type A (missing morning, where red points are artificial gaps), in which all data between sunrise and 10 a.m. in the evaluation set were held off. Data splits for the other two types of missing data (B and C) are shown in Figures S1.2 and S1.3 in the Supplementary Materials.

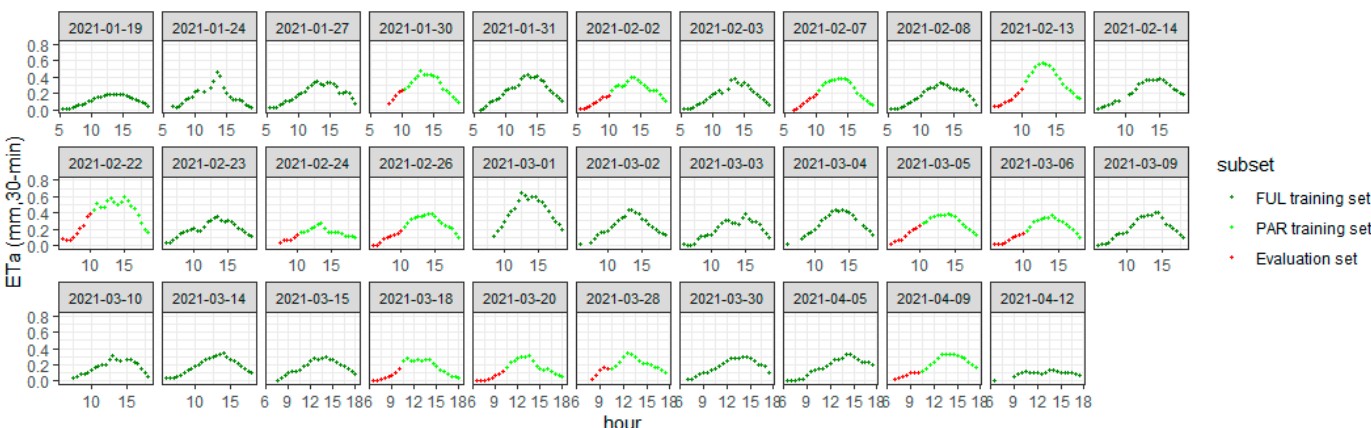

**Figure 5.** Split of the training and evaluation subsets to represent missing data Type A, i.e., missing morning.

### 2.3.3. Comparing the Model Infilling Performances

With the training and evaluation datasets created following Section 2.3.2, the four gap-filling models (three proposed models and MDV, see Section 2.2) were compared by their evaluation performance on infilling the missing 30-min $ET_a$ data. The root-mean-squared-error (RMSE) and the r-sqaured ($R^2$) were used to assess the model performance against the evaluation data; the former represents the average error in infilling the 30-min $ET_a$ relative to true observations and the latter represents the proportion of variance in infilling the 30-min $ET_a$ observations that can be explained by the model. As a further

reference to infilling performances, the RMSE values for the daily total $ET_a$ were also calculated for days that contain missing data.

## 3. Results

Figure 6 summarizes the RMSE of 30-min $ET_a$ for the four infilling models for the three types of missing data (A: missing morning; B: missing mid-day; C: missing afternoon), respectively. The MaxCor model consistently has the lowest model errors across all three situations, with RMSE values between 0.03 and 0.07. Following MaxCor, the next best models are the MDV (with RMSE ranging from 0.04 to 0.1) and the Sinusoidal (with RMSE ranging from 0.05 to 0.1) models, which have comparable magnitudes of errors. The Smoothing model has the worst performance with the highest errors for both Type A (missing morning, RMSE = 0.18) and Type C (missing afternoon, RMSE = 0.2). Considering the variance explained, the Sinusoidal and the MaxCor have higher ability to explain the observed variance, with $R^2$ values ranging from 0.7 to 0.87 and 0.68 to 0.82, respectively. The MDV model struggles to explain variance for Type B (missing mid-day, $R^2$ = 0.25) while the Smoothing model again shows limited performance for Types A and C (missing morning and afternoon, $R^2$ = 0.45 and 0.37, respectively).

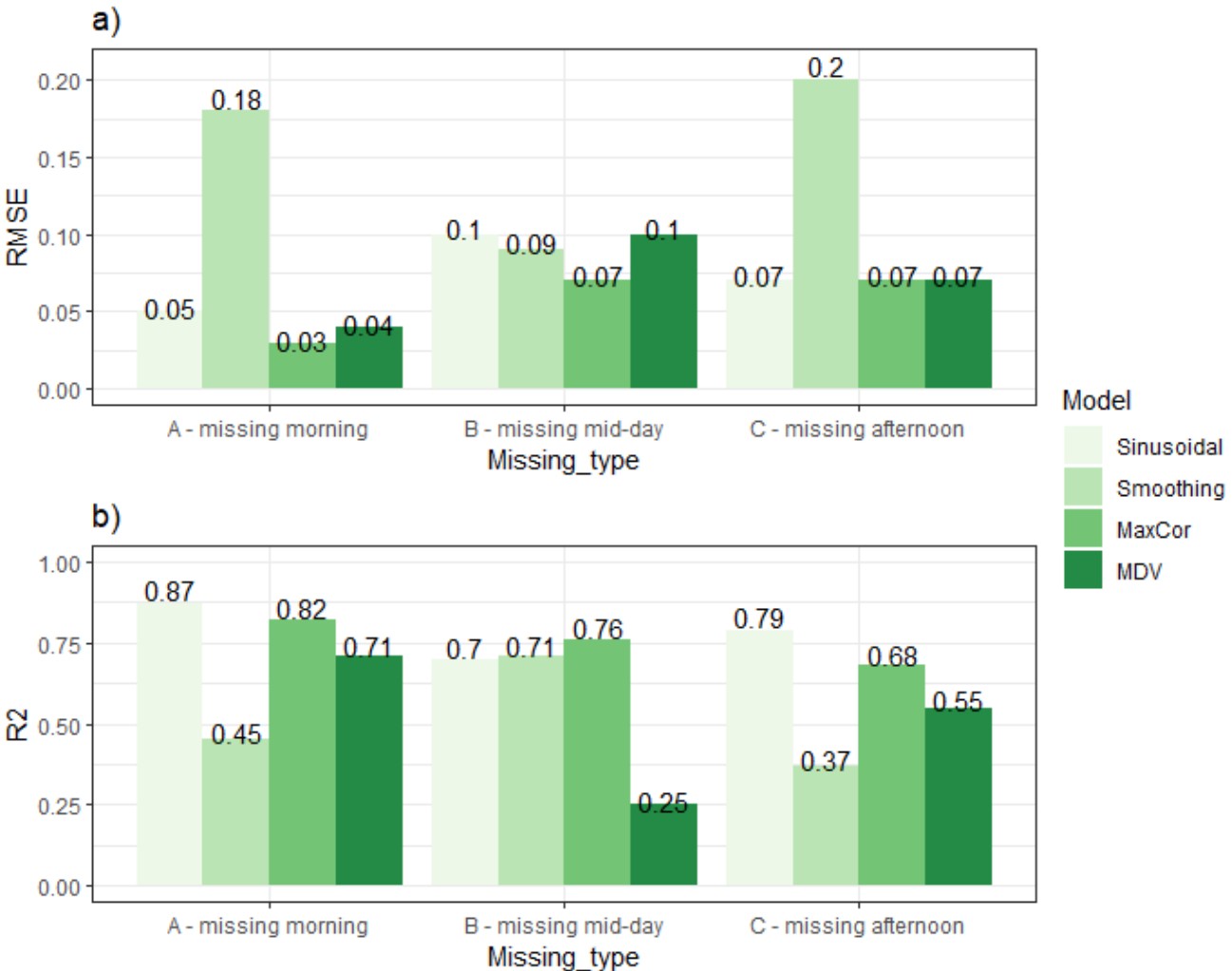

**Figure 6.** (**a**) RMSE and (**b**) $R^2$ of the 30-min $ET_a$ (in mm) for the infilled gaps, obtained from the four infilling models for each evaluation dataset that represents typical patterns of missing data: A—missing morning; B—missing mid-day; and C—missing afternoon.

With the above summary of performance of the four infilling models for the 30-min $ET_a$ data, we further aggregate infilling performance for days with missing data to understand

the expected accuracy at the scale of daily $ET_a$. Figure 7 summarizes the daily RMSE of four infilling models for the three types of missing data. The best-performing model, MaxCor, has mean errors of 0.26–0.62 mm in daily total $ET_a$.

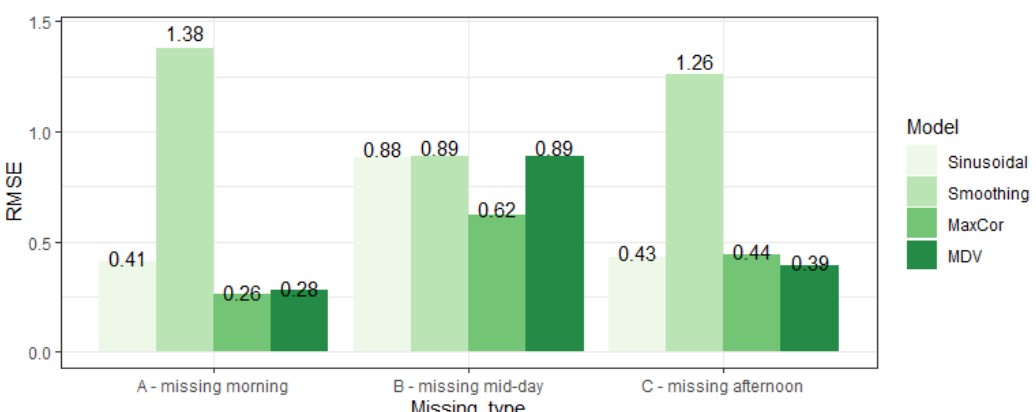

**Figure 7.** RMSE of daily $ET_a$ (in mm) for the infilled days with gaps, obtained from four infilling models for each evaluation dataset that represents typical patterns of missing data: A—missing morning; B—missing mid-day; and C—missing afternoon.

## 4. Discussion

### 4.1. Performances of Gap-Filling Models

Within the four infilling models that we evaluated, we see better performance for two models, which both perform gap filling for a day based on other 'similar' days and another model based on fitting functions to diurnal pattern of each day with gaps. Specifically, the MaxCor model identifies another day that has a complete record, while also having a similar diurnal pattern in the 30-min $ET_a$ to the existing records of the day to infill. The MDV model estimates missing records using the mean value of the specific time step in a day from neighboring days. The Sinusoidal model fits a sinusoidal function to the existing records within each day to infill and then uses the sinusoidal function for gap filling. The performances of all these three models are relatively stable across different missing data types, with less than 0.06 difference in the RMSE of 30-min $ET_a$ across different types of missing data.

In contrast, we see lower and more variable performances for the infilling model that uses smoothing functions to fill each day with gaps (Smoothing), where the maximum difference of RMSE of 30-min $ET_a$ across different missing data types is 0.11. This suggests a potential limitation of infilling performance due to the model structure. Specifically, the second-degree polynomial which is used as the smoothing function introduces more flexibility in the diurnal patterns of the 30-min $ET_a$ compared to either the MaxCor or the Sinusoidal models in which the diurnal pattern for the day to infill is bounded by either a similar temporal pattern in the actual records of the other day or by a sinusoidal function. Consequently, the fitted smoothing functions can be highly sensitive to fluctuations and outliers in the existing data, which can lead to spurious diurnal patterns and thus large errors in the infilled records.

There is no systematic pattern of how model performance varies across different types of missing data, suggesting that these variations are likely a result of individual model structures. For examples, the Sinusoidal and the MaxCor models show the worst performances for missing data Type B (missing mid-day), which may indicate the critical role of mid-day records. For the Sinusoidal model, these mid-day records generally consist of higher absolute values of $ET_a$ and thus have large impact on the calibration of the sinusoidal infilling function. For the MaxCor model, this low performance for missing mid-day records could be a result of the relatively high day-to-day variation of the temporal patterns in mid-day $ET_a$ (Figures 2 and 3), which leads to difficulties to infill missing records reliably using the temporal pattern within another day.

This study focuses on parsimonious gap-filling models for $ET_a$, which do not rely on any input variables other than $ET_a$ itself. Thus, they do not explicitly take into account any impact of weather conditions (e.g., solar radiation, cloudiness, rainfall, temperature), which are often considered important driving variables for ET. As such, all these parsimonious models share a common and natural caveat in maintaining robust performance on days when the influence of weather conditions on $ET_a$ is strong, and when weather conditions change abruptly. The infilling errors would be greatest when the data gap falls across a period of abrupt change in weather conditions. This is a fundamental limitation of all parsimonious gap-filling models that solely rely on $ET_a$ data.

To further understand the impact of this general limitation of parsimonious infilling models, we performed an additional analysis on the performance of the four gap-filling models under various cloud cover and solar conditions. Specifically, we plot the daily RMSE of each gap-filling model under the three types of data gaps against the daily ratio of actual solar radiation to clear-sky solar radiation (Figure S1.4 in the Supplementary Materials). We found that none of the four models are systematically influenced in performance by various cloud cover conditions within our dataset. This is likely due to the relatively limited variation in cloud cover conditions within our dataset to comprehensively characterize the effects of clouds on the accuracy of these gap filling approaches. Another plausible hypothesis is that these parsimonious gap-filling models, by considering the temporal patterns of sub-daily $ET_a$, have already effectively represented variation due to changes in cloud cover conditions (since solar radiation is an input when estimating $ET_a$ from EC-systems). However, this could only be the case if the cloud cover is relatively stable throughout the day; under situations where the amount of cloud cover is highly variable within a day, the diurnal variations of $ET_a$ would be much more difficult to be predicted from a simple smooth curve and/or averaging values from another day(s). This analysis illustrates the potential limitation of the infilling approaches under highly variable weather conditions, which is a general limitation to all parsimonious infilling models as discussed above. Similarly, we can expect much higher influence of the weather conditions on model performance when the weather is more highly variable throughout a day. Therefore, we strongly recommend individual investigation of this limitation when testing these (and potential other) parsimonious models to new datasets.

*4.2. Recommendations for Practical Situations with Different Data Availability*

In addition to the above comparison of model performances, we discuss the data requirement of individual infilling models to provide recommendations for different practical situations.

While the MaxCor and the MDV models are the best-performing models, both models also have higher data requirements for both days with complete data and the available data within each day to infill. Higher data availability on each day to infill enables a better understanding and thus a more reliable match of that day to the appropriate day with complete records. Large numbers of days with complete records are also critical for both the MaxCor and the MDV models: for the former, these provide a diverse set of diurnal patterns to match with the data in the day to infill; for the latter, more days with complete records can provide reliable mean estimates for each time step to infill.

Both the Sinusoidal and the Smoothing models use infilling functions that are fitted to the existing 30-min $ET_a$ records within the day with missing data. Therefore, neither require any day with complete data. Considering this together with the model performances in Section 3, the Sinusoidal model becomes the best choice to infill a dataset with limited complete days of record. An example for this situation is where the monitoring location experiences regular unfavorable wind direction that occurs for part of most days, leading to low-quality data (i.e., effective gaps) on most days of the eddy-covariance observations. Such a data quality issue is likely a result of inappropriate selection for the location of the eddy-covariance system, which may be due to practical constraints in many cases (e.g., to avoid conflict with machinery access to cropping fields, sites with naturally unfavorable

conditions in certain upwind directions). It is also worth noting that both the Sinusoidal and the Smoothing models do require a reasonable amount of data available in each day to infill, to enable a reliable infilling function to be developed. This data requirement is less strict for the Sinusoidal model, as the sinusoidal functions pose greater constraint on the diurnal patterns of 30-min $ET_a$. We encourage individual model evaluation in further case studies to obtain a specific and precise understanding of the impacts of data availability on model performance.

## 5. Conclusions

We adapted three parsimonious data-driven models to infill gaps in sub-daily $ET_a$ observations from eddy-covariance systems and evaluated these models together with another commonly used benchmarking model of similar data requirement. We applied these models to infill gaps in the 30-min $ET_a$ data collected from an eddy-covariance monitoring station installed in a maize field in southeastern Australia, over the 2020–21 summer season. We identified the best gap-filling model as a pattern-matching model to inform the diurnal pattern of the day to infill by another day with complete data (MaxCor). The second-best model is the benchmarking model, mean diurnal variation (MDV), closely followed by another proposed model which performs gap filling with sinusoidal functions fitted to the diurnal pattern of each day with gaps (Sinusoidal).

Further recommendations on model choice were made considering practical data availability. The best-performing MaxCor model relies on high data availability for both days with complete data and the available records within each day to infill. The Sinusoidal model does not rely on days with complete data while also offering reasonable performance, which makes it the best choice in situations where complete days of records are limited. We acknowledge that the performance of individual infilling models assessed may be specific to our study site and monitoring period, and results may differ across different climatic conditions, evaporative surfaces (i.e., crop type) and data availability. Therefore, local evaluation is highly recommended for future studies aiming to apply these infilling techniques. The strategies to allocate available records—to be used for calibration and evaluation of the infilling models and to be excluded from infilling due to data scarcity—should also be tailored for individual case studies, based on specific assessments of data availability (e.g., Figure S1.1). To facilitate further applications, we made the R codes of all four models evaluated in this study publicly available on GitHub https://github.com/DanluGuo/ETinfilling/blob/main/4_ETinfilling_Models_V2.R (accessed on 1 February 2022) along with the data we used to facilitate further applications.

**Supplementary Materials:** The following are available online at https://www.mdpi.com/article/10.3390/rs14051286/s1, Figure S1.1: Percentage 30-min ETa data availability within each day, sorted from the lowest to highest across the full monitoring dataset. Figure S1.2: Split of the training and evaluation subsets to represent missing data Types B i.e., missing mid-day.. Figure S1.3: Split of the training and evaluation subsets to represent missing data Types C i.e., missing afternoon. Figure S1.4: Daily RMSE of the four gap-filling models under the three typical patterns of missing data (A—missing morning; B—missing mid-day; and C—missing afternoon), plotted against the daily ratio of actual solar radiation to clear-sky solar radiation. Each panel shows one gap-filling model where the three missing data types are differentiated by colours. Table S1.1: Existing approaches to infill gaps in latent heat flux, carbon flux or directly for ETa. Orange cells highlight models that rely on additional input variable other than the variable to infill. Green cells highlight the only two existing parsimonious gap-filling models, the mean diurnal variation (MDV) and the analogue period (AP). Reference [26] is cited in the Supplementary Materials.

**Author Contributions:** Conceptualization, A.W.W., D.R., Q.J.W. and D.G.; methodology, D.G., A.W.W., D.R.; software, A.P., D.G.; validation, D.G., A.W.W., D.R., Q.J.W., A.P.; formal analysis, D.G.; investigation, D.G.; resources, A.W.W., D.R., Q.J.W.; data curation, A.P., D.G.; writing—original draft preparation, D.G.; writing—review and editing, D.G., A.P., A.W.W., D.R., Q.J.W.; visualization, D.G.; supervision, A.W.W., D.R., Q.J.W.; project administration, D.G.; funding acquisition, A.W.W., D.R., Q.J.W. All authors have read and agreed to the published version of the manuscript.

**Funding:** This research has been supported by the Australian Research Council via a Linkage Project (grant no. LP170100710), with contributions from our industrial collaborator, Rubicon Water.

**Institutional Review Board Statement:** Not applicable.

**Informed Consent Statement:** Not applicable.

**Data Availability Statement:** The authors have made all data used in this study publicly available at: https://github.com/DanluGuo/ETinfilling/blob/main/4_ETinfilling_Models_V2.R (accessed on 1 March 2022).

**Acknowledgments:** The authors would also like to thank Kevin Saillard, David Aughton, Emil Somers, Zitian Gao and Rodger Young for their assistance in the field monitoring campaign.

**Conflicts of Interest:** The authors declare no conflict of interest.

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
