# Peer review of "Parsimonious Gap-Filling Models for Sub-Daily Actual Evapotranspiration Observations from Eddy-Covariance Systems"

_remotesensing, doi:10.3390/rs14051286_

Round 1

Reviewer 1 Report

General Comments:

Although the paper is much improved, there is a quite a bit of analysis that is key understanding the different approaches and the conclusions of this study that are in the supplemental section. This information, much of which is referenced in the paper, should be moved to the main body of the paper. There are also a handful of relatively minor items that should be addressed for the benefit of the reader (see below).

Specific Comments:

  1. Line 66: Given that several of the references herein review techniques using neighboring measurements, interpolation/splining methods, and the like, what this seems to grossly understate the number of "parsimonious" approaches already available.

  1. Line 104: It would be helpful to include a brief discussion of how this approach works. The others are described below but no information is given regarding this paper.

3 Line 157: As evidenced by the reference to a figure in the supplementary section, there is a lot of important information in the supplemental section that should be integrated into the main body of the paper.

  1. Line 166: As noted above, it would be helpful to provide a brief overview of the MDV approach in this section as well for comparison purposes.

  1. Unnumbered 4 lines below Line 223: Actually, P has to be twice the day length. Otherwise, the position and magnitude of the infilled fluxes will offset.

  1. Line 392: This result remains questionable; it is likely that the variation in cloud cover conditions is too limited at this site to properly characterize the effects of clouds on the accuracy of these gap filling approaches. Moreover, the suggestion that a sinusoidal or quadratic fitting somehow accounts for the effects of cloud cover is only plausible if the amount of cloud cover does not change throughout the day. If the amount of cloud cover is highly variable, e.g., clear skies in the morning followed by cloudy or partly cloudy conditions in the afternoon, it is very unlikely to conform to a simple smooth curve.

Author Response

Please see our detailed responses to your comments and the corresponding revision in the attachment document.

Reviewer 2 Report

Review of the manuscript:

The presented study aims to develop and evaluate parsimonious approaches to fill gaps in evapotranspiration observations. The authors adapted three gap-filling models previously used for other meteorological variables. In the research was adopted a case study with field measurements from an eddy-convarience system over summer 2020-21, at a maize field in south eastern Australia. Authors find out as the best gap-filling MaxCor model. The scientific article has a standard structure with all necessary chapters including conclusions. I consider the conclusions in the manuscript are original. In this chapter are also further recommendations for the next research aiming to apply these infilling techniques. I did not find any serious defects in the work or in the presentation or ethical problems. Research in this area is relatively specific (perhaps for this reason, only 24 references are listed) and can have practical implications in relation to global warming and the growing problem of drought. In my opinion, the keywords are consistent with the content of the article. A short note is there is not necessary to repeat “actual ET”. In conclusion, I can state that after studying the article, I do not have any serious comments and manuscript “Parsimonious gap-filling models for sub-daily actual evapotranspiration observations from eddy-covariance systems“ meets the requirements and can be published in a Journal Remote Sensing.

Author Response

(The authors gave the same response as above.)

Reviewer 3 Report

I have no additional comments.

Author Response

(The authors gave the same response as above.)

Reviewer 4 Report

accept in the present form

Author Response

Please see our detailed responses to your comments and the corresponding revision in the attachment document.

This manuscript is a resubmission of an earlier submission. The following is a list of the peer review reports and author responses from that submission.

Round 1

Reviewer 1 Report

General Comments:

The manuscript by Gao et al. describes a comparison of four different gap-filling approaches for surface flux data. While the paper is well-organized and written, there is already a large body of large body of literature discussing different gap-filling and closely related interpolation techniques (Falge et al. 2001a, 2001b; Alvani et al. 2006; Alfieri et al. 2007; Moffat et al. 2007; Chen et al. 2012; Cammalleri et al. 2013; Park rt al. 2015; Goodrich et al. 2021), it is not clear what is novel here. For example, using reference ET to facilitate gap-filling was first suggested by Allen et al. (2007) and is often used to temporally upscale remote sensing estimates of ET (Colaizzi et al. 2006; Chavez et al. 2008; Alfieri et al. 2017). The authors need to better explain how this work is unique. They also need to provide more detailed information regarding their gap-filling approaches. The manuscript should be returned to the authors to address this issue as well as those described in the specific comments below.

Specific Comments:

  1. Line 33: The sentence beginning “ETa is of particular …” is awkwardly constructed and unclear. Are the authors suggesting that 99% of water loss from agricultural (crop) production is due to ET? Or that ET from crops accounts for 99% of evaporative water loss? In either case, that percentage seems very high.

  1. Line 39: The sentence beginning “These data gaps …” is awkward. It could be better expressed as “These data gaps limit the utility of these datasets when complete ETa records are necessary, such as for water balance calculations.”

  1. Line 143: “Julian day” is not correct here. The authors are referring to the “day of year” or “ordinal date”. The Julian date is the number of days since the beginning of the Julian period, i.e., 1 January 4713 BCE.

  1. Line 144: It is not obvious what is new or novel about these approaches. For example, a sinusoidal function has long been used to model incident solar radiation (Bird 1981), soil heat flux (Santanello and Friedl 2003) and other fluxes. A more complete explanation is needed to clarify this and to facilitate the use of these approaches by readers.

  1. Line 150: A sinusoidal function is only valid under clear-sky conditions. Under cloudy, and particularly under partially cloudy conditions, the fluxes are unlikely to vary smoothly.

  1. Line 156: More information is needed to explain this smoothing algorithm. Is it some type of shape-preserving spline? What is the mathematical form? How is it applied?

  1. Line 159: If the fluxes are missing or bad, it seems likely that variables needed to calculated the reference ET would also be missing. How is this dealt with?

  1. Line 163: The authors indicate that periods when ETa:ETo falls outside the range from 0 to 5 are excluded. What fraction of the data was excluded?

  1. Line 165: The description of Model 4 is confusing and needs to be revised for clarity.

  1. Line 204: How was this 70% threshold chosen? Having only 30% or 15 of 48 seem rather sparse to accurately gap-fill the data. Even if the data can be gap-filled, it would introduce substantial uncertainty to the dataset.

  1. Line 245: It is unclear why the models would perform so differently depending on the time of day when the gaps occurred. The authors should discuss this more completely.

  1. Figure 4: It would be helpful to normalize these values so the relative skill of the various approaches could be compared across time periods.

  1. Line 297: If much of the data can not be used because of issues related to wind direction, the location and configuration of the EC tower were poorly selected.

  1. Line 300: The phrase “require a reasonable amount of data” is vague. This needs to be clarified.

Author Response

Please see our responses in the attached document.

Reviewer 2 Report

Review of "Parsimonious gap-filling models for actual evapotranspiration observations from eddy-covariance systems"

General comments

This technical note presents four methods of gap-filling of evapotranspiration. The methodology and data processing is appropriate and well described, and the text is well written.

I have a question about method 4 on lines 170-173:

"The daytime ETa values in this ‘matching day’ are then used to derive the proportions of the 30-min daytime ETa to the daily total ETa. These proportions are then applied to the day to infill to estimate all 30-min daytime ETa while infilling the missing time steps"

Why calculate the proportions? How were they used to infill the PAR day? Why not fill the gaps in the PAR day with the respective records on the FUL matching day?

Specific comments

line, comment
173, missing period at the end
229, remove "to" or "of" in "... further reference to of ..."
260, "models"
262, are present

Figure 3 could follow the same design of Figure 2

Author Response

(The authors gave the same response as above.)

Reviewer 3 Report

Specific comments:

Line 26. Keywords. I suggest inserting maize field as a keyword. Looking at the Conclusion section, the authors state that the results are specific for this study area and to the maize crop. I think it's best to indicate the crop here.

Line 34. Reference [#3] Although this is a well-known paper, I suggest a more recent review on this topic.

Line 117. Please delete the abbreviation description, ETa already presented above.

Author Response

(The authors gave the same response as above.)

Round 2

Reviewer 1 Report

While the authors addressed many of the concerns discussed in the previous review, there are several aspects of the manuscript that still have not been discussed adequately. Disregarding the dubious claim that additional gap-filling is needed after other gap-filling approaches are employed, the authors provide two reasons why their methods are novel. First, he authors suggests their work is unique because most other approaches are intended for used with daily ET values, but this is untrue. While gap-fill is often applied at a daily time step for remote sensing application, this is because the daily time step is often the one of interest for practical applications. Even a cursory review of the extensive body of literature from both the micrometeorological and remote sensing communities (or just the examples cited in the previous review) shows that there are numerous statistical, empirical, and process-based gap-filling approaches that have been available for decades that can be applied at sub-daily timescales. Second, the authors point to the parsimony of their approaches, both the reference ET (Model 3) and correlation-based (Model 4) gap-filling approaches, which the authors point out have the greatest utility, are not new and require a suite of meteorological and other auxiliary measurements. Thus, the question remains, what is unique and advantageous about the methods the authors are proposing in this study?

Moreover, while Models 1 and 2 are truly parsimonious, they are problematic for other reasons. Both a simple sinusoidal (Model 1) or polynomial (Model 2) function is only truly applicable during the day under clear-sky conditions. Under overcast and especially partly cloudy conditions, it is highly unlikely that the fluxes will very smoothly. The authors add a short sentence acknowledging that this is a potential issue, but do not provide any meaningful discussion of the limitations of these approaches or how cloud cover might affect the gap-filling results. This needs to be addressed.

Until these concerns are fully addressed by the authors, the manuscript can not be recommended for publications. It should be returned to the authors to allow them the opportunity to incorporate thoughtful and detailed information into the paper the clearly demonstrates how their work is unique and of general utility to the readers of Remote Sensing.